# Behind the Gaps: A Narrative Review of Healthcare Barriers for Individuals with Serious Mental Illness

**DOI:** 10.3390/healthcare13192387

**Published:** 2025-09-23

**Authors:** Courtney N. Wiesepape, Sarah E. Queller Soza, Laura A. Faith

**Affiliations:** 1Department of Psychology, Indiana State University, Terre Haute, IN 47803, USA; 2Department of Psychiatry, Richard L. Roudebush VA Medical Center, Indianapolis, IN 46202, USA; sarah.quellersoza@va.gov (S.E.Q.S.); laura.faith@va.gov (L.A.F.)

**Keywords:** serious mental illness, mortality gap, healthcare, barriers, access, stigma, diagnostic overshadowing

## Abstract

Background: Individuals with serious mental illness (SMI) die significantly earlier and experience disproportionately higher rates of physical health issues compared with non-SMI groups. Despite advances in care, this mortality gap persists. One factor that contributes to this discrepancy is inadequate access to healthcare, as individuals with SMI are less likely to receive appropriate medical care. Methods: To better understand this, we completed a narrative review synthesizing existing literature on common barriers to care faced by the SMI community. We reviewed 34 articles and identified three primary barriers to receiving healthcare. Results: These included structural and logistical barriers (geographic location, access to technology and internet, disjointed medical and mental healthcare); intrapersonal- and patient-level barriers (symptoms and psychological impact of SMI, lack of awareness or prioritization of medical issues, medical mistrust, and limited health literacy); and provider- and system-level barriers (lack of knowledge or support for integrated care, lack of knowledge of SMI, stigma, and diagnostic overshadowing). Conclusions: We argue that addressing these issues requires a reorientation toward person-centered approaches that prioritize continuity, integration, and dignity in care for individuals with SMI, and we offer specific recommendations in service of these aims.

## 1. Introduction

Serious mental illness (SMI) refers to a mental, behavioral, or emotional disorder that significantly interferes with an individual’s ability to function and carry out major life activities [1,2]. SMI often includes formal diagnoses such as schizophrenia, bipolar disorder, and major depressive disorder, although there is variation in how SMI is defined and which diagnoses qualify for this distinction [1,3].

SMI is burdensome on an individual, system, and global level. People with SMI report that their illness interrupts their life in various ways, including impairing relationships and occupational functioning [4], which may limit their ability to contribute to society in productive ways. The economic burden of SMI is substantial and mounting, due to increases in prevalence, incidence, and disease burden [5]. For example, the estimated economic burden of schizophrenia in the United States doubled between 2013 and 2019, with its estimate being USD 343.2 billion in 2019 [6]. Relatedly, people with SMI have limited access to primary care, health screenings, and preventative services; thus, they tend to utilize more non-psychiatric services compared to healthy controls. For instance, a systematic review and meta-analysis found that compared to people without SMI, people with SMI are more likely to be admitted as nonpsychiatric inpatients, to be readmitted to the hospital, and to present to the emergency department [7].

People with SMI also have higher rates of health conditions and shortened lifespans compared to healthy controls. Overall, one study found that 87% of individuals with SMI reported at least one health comorbidity and 82% reported two or more physical health comorbidities [8]. A systematic review and meta-analysis found that people with SMI are more than twice as likely to die from infectious diseases and more than three times as likely to die from respiratory infections [9]. Another review found that mortality among people with schizophrenia is related to several health-related or modifiable factors, including suicide, disease (e.g., pneumonia, infectious disease, diabetes, and others), or a comorbid substance use disorder [10].

While antipsychotic use (especially second-generation antipsychotics) has been found to be protective against mortality, a large body of research shows that nonadherence to medication is common for people with SMI [11] and, for medication adherent individuals, the use of antipsychotics increases the risk of metabolic syndrome and associated health complications. Use of antipsychotics have been linked to weight gain, diabetes, hypertension, and dyslipidemia, all of which are associated with increased risk of a cardiovascular event and other negative health outcomes [12]. Metabolic syndrome is a common outcome of antipsychotic usage and is generally thought to include weight gain, hypertriglyceridemia, and increased insulin, glucose, and low-density lipoprotein cholesterol levels. One study found that risk of heart disease and stroke tripled in individuals with metabolic syndrome [13]. Although physical health and metabolic monitoring is recommended for individuals taking antipsychotics, this practice remains suboptimal, with most hospitals not reporting any metabolic monitoring practices for individuals diagnosed with SMI and prescribed antipsychotics [14].

A separate study found that the majority of deaths in individuals with SMI are attributable to physical diseases, with cardiovascular diseases accounting for the greatest percentage of deaths [15]. Obesity and other metabolic abnormalities represent significant risk factors for cardiovascular disease, and these abnormalities are especially prevalent within the SMI population due to a number of factors, including physical inactivity, poor diet, and side effects of psychotropic medications [15]. The rate of obesity in individuals with SMI is twice the rate of the general population, which, in addition to being a risk factor for cardiovascular disease, puts individuals with SMI at risk for developing type 2 diabetes and cancer and reduces life expectancy by 15 years [16]. Other conditions that are significantly more prevalent among individuals with SMI than in the general population include liver disease (with a rate nearly six times the general population), diabetes and chronic bronchitis (nearly three times as high) and viral hepatitis (more than twice as high) [8]. Additionally, rates of smoking are much higher in individuals with SMI, which further contributes to physical health comorbidities.

It is important to underscore the bidirectional relationship between physical and mental health in SMI. Notably, individuals with SMI are almost twice as likely to experience physical multi-morbidities compared to the general population, highlighting how poor physical health can exacerbate psychiatric symptoms and vice versa [17]. This connection demonstrates the importance of considering barriers to both physical healthcare and mental healthcare in SMI and the need for interventions that consider all aspects of health. However, despite advances in medical care, the mortality gap in SMI persists and may be widening [18]. One factor that may contribute to this is inadequate access to healthcare in SMI populations. Individuals with SMI have lower access to primary and tertiary healthcare services, leading to increased usage of emergency services [19,20]. They are less likely to receive appropriate treatment after significant medical events (e.g., stroke), have lower rates of surgical intervention, receive substandard care for chronic conditions (e.g., diabetes), and are less likely to undergo routine cancer screenings [21].

### Current Aims

To deepen our understanding of the persistent mortality gap and the role of inadequate healthcare in perpetuating it, we conducted a narrative review of the barriers to healthcare that individuals with SMI face. We aimed to synthesize recent literature on barriers to care, identify individual-, provider-, and system-level challenges to accessing care, and examine how current healthcare systems inadequately support and meet the health needs of individuals with SMI. This narrative review aims to produce a timely update of barriers to care faced by individuals with SMI, especially given widespread changes to the healthcare system as a result of the COVID-19 pandemic and the rapid expansion of telehealth services. We reviewed barriers to accessing both physical and mental healthcare, as mental health cannot be separated from physical health and vice versa. Broadly, we sought to highlight gaps in knowledge and practice that contribute to ongoing disparities and to inform future clinical, research, and policy efforts in a way that is both practical and relevant for stakeholders. We chose a narrative review for its flexibility and ability to explore the complexities of the mortality gap in SMI in order to form hypotheses that may be useful for future research and practice.

## 2. Materials and Methods

Given the focus of this narrative review on healthcare access for individuals with SMI, PubMed was selected as the primary database for our literature search due to its coverage of biomedical, psychiatric, and health services research and relevance to both physical and mental healthcare. Of note, the purpose of this narrative review was not to provide a systematic or exhaustive synthesis but rather to highlight persistent gaps in care and to identify key themes in the existing literature within the last five years. To ensure rigor and transparency in this narrative review, we followed recommendations from the Scale for the Assessment of Narrative Review Articles (SANRA), which emphasizes clear justification of the review, explicit aims, transparent literature search strategies, appropriate referencing, critical synthesis of findings, and clear presentation of data [22]. While SANRA is not a tool for formal quality appraisal, it guided our approach to synthesizing and reporting the literature in a structured and methodologically transparent way.

Search terms were developed to focus on both mental healthcare and physical healthcare access for individuals with SMI. Two separate searches were conducted. The first search combined terms for SMI with terms related to mental healthcare access and systematic barriers. The final search string for the mental health search was: (“serious mental illness” OR “severe mental illness” OR “SMI” OR schizophrenia OR schizoaffective OR psychosis) AND (“mental health care” OR “psychiatric treatment” OR “behavioral health”) AND (access OR barriers OR stigma OR disparities OR inequities), limited to publications from the last five years and indexed in PubMed. This search yielded 512 results. A second search combined the same SMI terms with those specific to physical health care access and the same barrier-related terms, yielding 378 results. The final search string for the physical health search was: (“serious mental illness” OR “severe mental illness” OR “SMI” OR schizophrenia OR schizoaffective OR psychosis) AND (“physical health care” OR “primary care” OR “medical care”) AND (access OR barriers OR stigma OR disparities OR inequities), limited to publications from the previous five years and indexed in PubMed. The article search was conducted between 16 July and 25 July 2025.

Articles were reviewed in two phases for relevance to the overarching question of barriers to access to care for individuals with SMI (see Figure 1). Other inclusion criteria included being published in English and within the last five years. We restricted inclusion to the past five years to capture the most recent evidence, with particular attention to studies conducted during and after the substantial shifts in healthcare delivery brought about by COVID-19, as these changes may have altered the nature of barriers faced by individuals with SMI. Exclusion criteria included being published outside the selected timeframe, not focused on an SMI population or access to healthcare, and not peer reviewed. The first review included screening titles and abstracts for inclusion and exclusion criteria. The majority of articles were removed for lack of relevance to healthcare access or not being focused on SMI. After this initial search, 99 articles remained from the mental health search and 53 articles remained for the physical health search. At this point, the two searches were combined due to substantial conceptual and thematic overlap across the results, resulting in a combined 152 articles. Seven articles were removed due to being duplicated across searches. Next, a full-text review was conducted for the remaining 145 articles. A substantial number of articles (*n* = 111) were excluded at the full-text review stage. This reflected our decision to adopt an intentionally broad approach during the initial screening phase to minimize the risk of missing relevant studies. Many of the excluded articles did not specifically address healthcare barriers, focused on interventions to decrease barriers, or inadequately covered SMI populations. While this conservative approach increased the volume of exclusions, it ensured that the studies included in the final synthesis aligned closely with our aims. This narrative review focused on primary studies; however, selected review articles are also included to provide context, summarize broader trends, and highlight areas of consensus or gaps in the literature. This process resulted in 34 articles that were included in the final synthesis, discussed in Section 3.

## 3. Results

Our review of the literature revealed three primary themes related to the barriers that individuals with SMI face when trying to access or engage in healthcare: structural and logistical barriers, intrapersonal or patient-level barriers, and provider- or-system level barriers. We discuss each theme and relevant subthemes below.

### 3.1. Structural and Logistic Barriers

This theme describes the structural and logistical barriers to accessing adequate healthcare that individuals with SMI face, including how healthcare systems fail to reach people with SMI and are more difficult to access for individuals with SMI.

The geographic locations of clinics, hospitals, and other health centers can create significant barriers to care for individuals with SMI. Proximity to mental health services is one key factor, indicating that inaccessibility or restricted access to healthcare based on location contributes to inadequate care. Travel time is significantly associated with the number of appointments scheduled and attended and program status (e.g., graduated, active, disengaged). Specifically, greater travel time has been associated with lower likelihood of graduation for Hispanic service users and a higher rate of disengagement for non-white service users [23], highlighting important implications of structural and geographic barriers for service users of color and individuals from disadvantaged backgrounds. In addition, individuals who live in more rural areas, and thus experience further proximity from major medical centers and other healthcare facilities, face additional barriers to accessing adequate mental and medical health services [24]. Geographical location or proximity to care can also increase ease of care; for example, the convenience of a COVID-19 vaccination location was associated with increased vaccine uptake among individuals with SMI [25], suggesting that if a location is inconvenient, individuals with SMI may be less likely to access vaccinations and other healthcare. This finding aligns with the research discussed above that reports proximity to care as a vital factor to accessing appropriate care in SMI. In addition to the physical location of services, lack of or limited transportation represents an important barrier to accessing care. Without consistent and reliable private or public transportation, even minor distance challenges become significant barriers. This may be especially true for individuals living in rural areas. For example, difficulty with community mobility (e.g., the ability to safely and independently move around one’s community using various forms of transportation) and poverty have been identified as barriers to engagement with healthcare by stakeholders in rural areas [26]. Taken together, distance from care locations and lack of transportation represent significant barriers for individuals with SMI in accessing healthcare.

Telehealth is becoming increasingly common and may serve as a solution to geographical and transportation barriers; however, engaging in telehealth requires access to a device (e.g., smartphone, computer) and reliable internet. Research suggests that individuals with SMI may lack technological skills and face difficulty in consistently accessing these resources, termed the “digital divide” [27]. Individuals with SMI often struggle to maintain consistent internet access even though most own smartphones. In addition, phones that are broken, lost, or uncharged result in additional disruptions to care [27]. Importantly, disruptions in access to technology that allows for telehealth may be especially prominent in individuals with SMI experiencing homelessness or poverty. Some programs have provided individuals with SMI smartphones (e.g., certain Medicaid programs) to overcome access to technology as barriers to care. However, barriers such as monthly data limits and difficulty with account management persisted [28]. Although digital mental health interventions improve access and support recovery for individuals with SMI, simple access to a smartphone is not sufficient. Even with access to a smartphone, additional barriers to telehealth services must be considered, including low digital literacy (e.g., having a limited understanding of devices, being unfamiliar with applications and notifications) and financial burden (e.g., being unable to consistently pay phone bills) [29]. Thus, although the expanding use of telehealth can mitigate some barriers, including geographical and transportation barriers, it simultaneously presents new obstacles for individuals with SMI.

In addition to geographical and telehealth-related barriers, fragmented and uncoordinated care is another structural barrier individuals with SMI face that often results in poorer access and outcomes. Individuals with SMI and family members have described a lack of communication and coordination between medical providers and mental health providers, leading to fragmented care [30]. The separation of medical and mental healthcare has also been identified as a barrier to treatment and a reason for people with SMI falling through the cracks [31]. Importantly, this barrier exists broadly (e.g., fragmentation or lack of coordination between mental and physical healthcare systems) and within specific domains of healthcare. For example, specific to oral health, lack of coordinated care and difficulty navigating dental systems have been identified as significant barriers to receiving dental care [32]. Similarly, specific to cardiovascular risk, general practitioners reported that barriers to management of risk included poor communication and information exchange between providers (i.e., psychiatrist and general practitioners) [33].

Together, structural and logistical barriers include geographical distance from care, challenges to engaging in telehealth, and fragmentation or lack of coordination of healthcare systems. Distance from care is often compounded by lack of access to consistent and reliable transportation. Although telehealth has been touted as a solution to geographic barriers, wide implementation of telehealth comes with unique challenges for individuals with SMI. Finally, fragmented healthcare systems create barriers of their own while worsening existing barriers (e.g., requiring individuals with SMI to attend and navigate mental and physical health appointments in different locations).

### 3.2. Intrapersonal and Patient-Level Barriers

Intrapersonal and patient-level barriers reflect how an individual’s perceptions of oneself, others, and the world impact attitudes towards engaging with the healthcare system. For example, individuals with SMI may face barriers to engagement based on the severity of their psychiatric symptoms, a prioritization of mental health over physical health, or a distrust of healthcare systems.

One class of barriers includes how mental health symptoms associated with SMI (e.g., paranoia, difficulty with memory or concentration, disorganization, etc.) impact one’s ability to engage with healthcare providers and systems. For example, individuals with SMI and their caregivers have cited intrapersonal barriers to accessing care, including cognitive capacity and the impact of their mental illness. Specifically, individuals with SMI reported that acute symptoms, such as disorganization, made it more difficult to engage in care. Other barriers included challenges with scheduling, organizing, or remembering appointments, anxiety about using public transportation or leaving home, and general difficulties with engagement [34]. Psychiatric symptom severity has also been associated with delays in seeking medical care in multiple studies for people with SMI, reflecting the bidirectional nature of physical and mental health. Specifically, delayed or impeded care has been found to be related to paranoia and depression symptoms [35], as well as motivation and cognitive difficulties [31]. In addition, psychiatric symptoms may result in delayed or missed preventative care screenings [36]. Together, these findings highlight the importance of addressing both physical and mental health (and their interaction) when attempting to reduce barriers to care for individuals with SMI.

Related to symptoms interfering with obtaining adequate healthcare, it has also been suggested that patient prioritization of mental health concerns over medical concerns can limit medical healthcare. This finding emerged in numerous studies reviewed. For example, prioritization of mental health and substance use concerns over medical care coupled with other patient level factors (e.g., homelessness) were identified as barriers to accessing primary care by mental health providers [37]. Relatedly, medical providers have reported focus on other patient concerns as a reason for lower levels of cancer screening in individuals with SMI [38]. These findings indicate that both mental health and medical providers identify a lack of priority for physical health as a barrier to adequate care. Importantly, only one study supported this view from the patient perspective [36]. In contrast to a lack of prioritization of medical care, some studies identified lack of awareness of physical health problems as a barrier to medical treatment in SMI populations [39,40]. Regardless of whether it stems from limited awareness or low prioritization, a lack of attention to medical issues represents a significant barrier to accessing medical healthcare.

Distrust in the medical system, or medical mistrust, has also been cited as a barrier to individuals with SMI seeking and receiving adequate healthcare by providers [31] and patients [36]. In one study, over one-third of African Americans with SMI demonstrated elevated medical mistrust, with authors suggesting that this influences willingness to seek care [41]. Of note, it is important to recognize that medical mistrust among individuals with SMI may be related to multiple, interacting factors. Mistrust may be related to symptomology associated with SMI, such as paranoia. However, it may also reflect a history of stigma, discrimination, systemic neglect, or direct experiences of medical trauma and mistreatment. At the intrapersonal level, these experiences often intersect with cultural factors, stigma, and discrimination, compounding barriers to care and shaping how individuals may perceive and engage with treatment.

Finally, limited health literacy in individuals with SMI presents an additional significant barrier to healthcare. Healthcare systems often demand a high level of literacy to understand and act on documented information (e.g., healthcare forms, informed consents, etc.). These literacy demands were found to often exceed the literacy skills of individuals diagnosed with SMI in shelters, and this mismatch was greater in the SMI population than in the general population [42]. Similar findings have been found in a clinical high-risk group with low mental health literacy and illness stigma identified as barriers to obtaining treatment [43]. Of note, health literacy extends beyond the ability to read and comprehend complex healthcare documentation and information. It also encompasses functional gaps, such as uncertainty or confusion about when to seek services, which has been identified among individuals with SMI as a barrier to accessing care [44]. Increasing health literacy in SMI or lowering literacy needs to engage in healthcare systems may be required to begin to address these barriers.

At the patient-level, individuals with SMI face multiple, interacting factors. These include psychiatric symptoms that may impede access to care, a lack of prioritization or awareness of medical needs, distrust in the medical system, and limited health literacy at both a practical and functional level. These barriers are difficult to disentangle given their interactive nature (e.g., the bidirectional nature of physical and mental health problems), suggesting that interventions may need take a more holistic and integrative approach.

### 3.3. Interpersonal and Provider-Level Barriers

This theme focuses on how negative or inconsistent provider experiences can discourage ongoing patient engagement. This may include provider stigma, lack of knowledge/training, or other provider-level barriers.

One contributing barrier to accessing medical care for individuals with SMI is lack of provider knowledge and support for integrated care. Broadly, lack of sufficient training and experience to identify medical problems in people with SMI and support them in accessing treatment have been identified as barriers in psychiatric providers [39]. Although mental health providers tend to report awareness of the importance of physical health in SMI, they also report numerous barriers to providing or encouraging medical care, including uncertainty about the role of mental health in physical health checks, a lack of medical knowledge and training, resource limitations [45], and negative attitudes about this approach [46]. Importantly, provider barriers persist even in programs designed for people with SMI. For instance, Assertive Community Treatment (ACT) team members identified lack of staffing and a limited number of medical providers who are willing to work with people with SMI as significant barriers to treating physical health concerns for people with SMI within ACT [40].

As discussed above, lack of knowledge about medical care and considerations in mental health providers has been identified as a barrier to individuals with SMI accessing care. Correspondingly, lack of knowledge about SMI in medical providers has also been recognized in studies as a barrier to providing adequate care to individuals with SMI. Generally, physicians do not feel prepared to treat SMI [47]. For example, in a study of family physicians, 25% were unaware of early psychosis intervention services in their area and 80% preferred to work with a specialist to manage the health of their patients with psychosis [48]. This lack of knowledge about and comfort with working with SMI has been identified as a primary driver of poor outcomes. Importantly, these findings remain consistent in psychiatrists (e.g., medical doctors with mental health specialty). For example, although psychiatrists recognize the importance of addressing health needs, they report various barriers including lack of training or knowledge, discomfort, and limited time [49].

In sum, mental health professionals frequently report limited knowledge of medical and physical health, while medical professionals similarly report limited knowledge of mental health, highlighting a bidirectional gap in interdisciplinary expertise. Both lack of initial training and continuing education likely contributes to this gap. For example, lack of continuing education in mental health was identified as a barrier to care and connected to lower confidence in providing care to individuals with SMI by healthcare providers [31]. These findings also apply to specific interactions between physical and mental health. Specific to physical health (i.e., cardiovascular risk), general practitioners reported that barriers to management of risk included skepticism or doubt about patient compliance and misconceptions about the health of individuals with SMI [33]. Correspondingly, in psychiatry, gaps in competency in the ability to adequately manage antipsychotics due to a lack of postgraduate training and continuing education have been identified as primary barriers to completing medication reviews [50]. Taken together, lack of knowledge or expertise impacts various levels of physical and mental healthcare.

Individuals with SMI are a highly stigmatized group, and this stigma extends to healthcare providers, which serves as a provider-level barrier to adequate care [31] and has been identified as a primary driver of poor health outcomes in SMI [47]. As a caveat, stigma likely impacts numerous other barriers discussed here due to its pervasive impact in healthcare systems. Stigma in healthcare impacts various groups with SMI, including mothers [51], and may interact with other aspects of identity to increase experiences of discrimination and further limit access to healthcare. Additional barriers related to medication reviews are also related to stigma toward individuals with SMI. Barriers to performing medication reviews by primary care providers include low expectations for recovery (e.g., believing that people with SMI do not improve or recover), the assumption that individuals with SMI are unable to understand or actively participate in reviews [52], and fears of a “catastrophic event” if a patient stops antipsychotics [50]. In addition, collaboration between healthcare professionals and individuals with SMI has been described as “troublesome” primarily due to stigma and prejudice related to SMI [39]. More broadly, physicians tend to endorse stigmatizing beliefs about SMI, including that people with schizophrenia are dangerous and unpredictable, as well as actions based on stigma, including a desire for greater social distance toward individuals with schizophrenia [53].

One way that stigma impacts medical care for individuals with SMI is through diagnostic overshadowing. This is defined as the increased likelihood of physical health problems and symptoms being attributed to an existing SMI diagnosis or other mental health factor and subsequent referral to specialty mental healthcare in place of needed medical intervention [53]. In summary, an individual’s SMI diagnosis takes precedence over physical health concerns or symptoms and prevents them from obtaining appropriate care. Individuals with SMI and their family members report facing stigma and discrimination from primary care providers and describe their physical healthcare being neglected due to diagnostic overshadowing, leading to them feeling dismissed and discouraged from seeking medical treatment [30,54]. Furthermore, in a survey of individuals with a mental health condition seeking primary healthcare services, 20% of participants reported diagnostic overshadowing and 10% reported discrimination based on their mental health condition. Importantly, individuals with SMI had significantly worse experiences across all measures of quality of care [55]. The impact of diagnostic overshadowing plays a role not only in primary healthcare but also in specialty medical care. For example, symptoms of COPD (e.g., shortness of breath) are often viewed as psychosomatic in SMI, which leads to underdiagnosis and thus less treatment and worse outcomes [56]. This is especially important given the high rates of smoking and respiratory illnesses found in people with SMI.

Barriers to accessing adequate care at the provider level include lack of knowledge among medical and mental health providers about each other’s domains and the intersection of these domains, a gap that is often exacerbated by a lack of continuing education. In addition, provider stigma is a prominent barrier to quality care in SMI. Because stigma and diagnostic overshadowing are interacting barriers to care at the provider-level, they are often found together in practice and may be addressed in tandem. See Table 1 for a summary of results and Table 2 for a summary of themes.

## 4. Discussion

People with SMI experience significantly worse health outcomes compared to the general population, including higher rates of chronic physical conditions, reduced life expectancy, and greater difficulty accessing and engaging with healthcare services. We completed a narrative review to explore how current healthcare systems fail people diagnosed with SMI, with a specific focus on the barriers individuals with SMI face when accessing adequate healthcare. Our review revealed three main themes: (1) structural and logistical barriers, (2) intrapersonal and patient-level barriers, and (3) provider and systemic barriers. Structural and logistical barriers included geographic locations of clinics, hospitals, and healthcare systems, reduced access to technology and internet, and disjointed mental and medical healthcare systems. Patient-level barriers included symptoms and psychological impacts of SMI, lack of awareness or prioritization of medical issues, medical mistrust, and limited health literacy. Finally, provider- and system-level barriers included a lack of knowledge and support for integrated care among providers, lack of knowledge about SMI, stigma toward individuals with SMI, and diagnostic overshadowing.

Other studies have utilized a similar barrier organizational structure. For example, one recent scoping review of diabetes care in SMI found that factors contributing to inadequate care can be found at the patient-, provider-, and healthcare system levels [57]. Similarly, another review identified barriers to accessing services at the service user, professional, and institutional bureaucracy levels but noted a lack of consideration of the service user perspective [58]. The current narrative review identified a similar barrier structure (i.e., structural/logistical barriers, patient-level barriers, and provider/organization level barriers) in access to physical and mental healthcare within SMI and utilized qualitative studies to ensure patient and provider perspectives were incorporated. Although these barriers are distinct, there is some overlap between levels. For example, stigma plays a role in both the patient- and provider-levels. At the patient level, self-stigma (e.g., societal stigma directed inward) is the primary barrier, whereas at the provider level, societal stigma is the primary barrier (e.g., stereotypes, prejudice, and discrimination held by providers as members of society) [59]. Barriers that exist in more than one barrier category may offer unique opportunities for intervention to create meaningful change.

Of note, several factors commonly assumed to contribute to healthcare barriers did not prominently emerge in our review. For example, issues such as financial or insurance challenges, cultural influences, and trauma appeared less frequently than expected. While some studies did touch on these areas (e.g., the role of ethnicity in first-episode psychosis treatment engagement), it is likely that a broader body of literature beyond our review addresses these important considerations. Other social determinants of health (e.g., socioeconomic status, education) likely influence healthcare access for individuals with SMI, potentially exacerbating disparities due to multiple forms of disadvantage. While this review briefly considers these factors (e.g., the role of discrimination in medical mistrust in SMI), further attention is needed to fully understand how structural-, patient-, provider-, and system-level barriers interact, particularly in the context of cultural influences and broader social determinants of health. For example, stigma directed toward SMI may be worsened by other types of discrimination due to race, ethnicity, economic status, etc. Additional research focused on culture, intersectionality, and social determinants of health is needed.

In addition, much of the research reviewed here was completed in Western and developed countries. Other studies have more directly focused on rural and non-Western countries. For example, one qualitative meta-synthesis focused specifically on barriers and facilitators to seeking medical help in rural patients with mental illness. This study identified three themes, including (1) navigating vulnerability and empowerment (individual level), (2) navigating the external environment, and (3) connectivity within the healthcare ecosystem (health service level) [60]. In a follow up editorial, authors note that rural communities in China face additional challenges similar to those addressed in this review, including lack of specialized psychiatric facilities (geographic limitations), difficulty with medication distribution, and lack of specialty providers [61]. Additionally, time and resource constraints are thought to disproportionally impact rural healthcare systems, which may widen barriers and disparities for individuals with SMI living in rural areas [62]. Additional research that focuses on less developed, rural, and non-Western countries is needed to better consider the influence of other healthcare systems and cultural beliefs on barriers to healthcare in SMI.

Overall, this narrative review highlights important findings related to the barriers to care faced by individuals with SMI. These findings inform the following recommendations that aim to improve care for individuals with serious mental illness.

### Recommendations

A range of interventions and policies have been implemented to decrease barriers to healthcare for individuals with SMI. For example, integrated care models designed to combine mental and primary care services and telehealth services are commonly cited as methods to improve the mortality gap in SMI. Research exploring whether these interventions are effective is comparatively limited; however, one recent scoping review noted that integrated care models were associated with increases in primary care access for individuals with SMI [63]. There is currently mixed support for the use of telehealth in SMI, with some studies reporting that a high proportion of telehealth visits may reduce access and continuity of physical and mental healthcare [64], while others have reported greater telehealth use is associated with modest increases in contact with healthcare professionals, especially in rural communities [65]. Ultimately, more research into interventions to increase access to mental and physical healthcare in SMI is needed. Our primary recommendations based on the results of this narrative review include (1) improving access to resources for telehealth, (2) promoting healthcare integration, (3) utilizing shared decision-making to improve health literacy and decrease medical mistrust, and (4) encouraging ongoing SMI education and stigma reduction interventions.

Telehealth has improved access to specialty care for individuals with SMI, growing 425% between 2010 and 2017 for individuals diagnosed with schizophrenia or bipolar disorder in rural areas [66]. This number likely increased even more during the COVID-19 pandemic, when providers and agencies responded to national emergencies and quarantines by offering telehealth more regularly. However, access to care is still limited due to geographical locations of clinics and challenges related to technology and reliable internet. Access to care for individuals with SMI could likely be improved by ensuring individuals have a working smartphone, reliable internet, and adequate digital literacy. Increasing the availability of affordable smartphones and data plans could enable more individuals with SMI to connect with providers remotely. Additionally, initiatives aimed at enhancing digital skills, including programs focused on developing digital skills, empower patients to effectively use telehealth platforms and improve access to care [67]. Thus, increasing the availability and competency of telehealth programs while offering individualized support to individuals with SMI could greatly improve structural and logistical barriers to care.

Greater integration of mental health and medical services is a common recommendation to improve health outcomes in SMI [30,63], and individuals with SMI generally support this recommendation [68,69]. Medicaid health homes represent one recent attempt to move toward integration. Health homes were designed to coordinate care for individuals with chronic medical issues, integrating primary, acute, and behavioral health resources. Recent studies show that these increase service use [70] and improve health outcomes for individuals with SMI, specifically [71]. One large-scale study called the Recovery After an Initial Schizophrenia Episode (RAISE) project aimed to promote coordinated specialty care to reduce the duration of untreated psychosis after an initial episode [72,73] and showed positive results [74,75]. Although integration of mental and medical healthcare has been shown to improve outcomes and enhance care coordination, its implementation can be difficult due to various barriers, including limited funding, fragmentation of existing systems, and lack of provider training.

Shared decision-making may be one way to address patient-related barriers related to accessing care, including limited health literacy, medical mistrust, and difficulty understanding and/or prioritizing medical issues. Shared decision-making is defined as a collaborative approach to healthcare that involves patients and providers working together to discuss healthcare options and make healthcare decisions in a way that centers the patient [76]. One way to encourage shared decision-making is to have training for providers that includes didactic information about shared decision-making and experiential training on how to practice shared decision-making with patients. Although patients with low health literacy struggle to engage in shared decision-making, they still prefer this approach [77], and this may ultimately improve health literacy. Additionally, by definition, shared decision-making helps individuals to better understand, discuss, and prioritize their healthcare goals with the input of their provider, potentially leading to increased awareness and prioritization of medical issues. Finally, shared decision-making has the ability to increase trust in providers [78]. Although there are additional barriers to successful implementation of shared decision-making [79,80], it has the potential to improve health outcomes in SMI by decreasing the impact of numerous patient-level barriers. For example, utilizing shared decision making can enhance self-management in SMI, which in turn has been shown to improve functioning, quality of life, and overall outcomes [81]. More research is needed to empirically evaluate shared decision-making approaches to address these barriers.

Finally, encouraging ongoing education about SMI for providers and demonstrating interventions designed to reduce provider stigma could likely reduce provider-level barriers to care for individuals with SMI. Increasing healthcare providers’ knowledge about SMI can lead to greater comfort and confidence in treating this population and improved clinical competencies. Interventions aimed at increasing SMI knowledge may include required continuing education or supervision and consultation. Interventions that target stigma are also important for improving access to care for individuals with SMI. Healthcare providers generally support anti-stigma interventions aimed towards primary care and specialty providers [82]. These interventions typically fall into two main categories: education-based and contact-based approaches. Education-based interventions aim to increase knowledge about mental illness (e.g., challenging misconceptions, increasing understanding of symptoms), while contact interventions involve direct exposure to individuals with lived experience of SMI. Studies show that these interventions, which include education-based [83,84] and contact-based [85,86] interventions, are effective in reducing stigma. Lack of knowledge of SMI and stigma toward SMI, along with the related concept of diagnostic overshadowing, are major barriers that may be reduced with appropriate interventions to increase knowledge and decrease stigma.

## 5. Conclusions

In summary, and based on the barriers reviewed here, we recommend a reorientation toward person-centered approaches that prioritize continuity, integration, and dignity in care for individuals with SMI. For policy makers, this means supporting integrated care models that bring together medical and mental health services, expanding reimbursement for individualized telehealth, and funding stigma-reduction initiatives at the system level. For clinicians, this includes adopting shared decision-making practices, offering both in-person and telehealth options to maximize accessibility, and engaging in ongoing training to reduce stigma in practice. For researchers, we recommend continued evaluation of integrated and telehealth models, as well as the development of scalable stigma-reduction and shared decision-making interventions. Finally, we emphasize that addressing SMI is a shared responsibility, and broad, interdisciplinary training is essential to improving health outcomes.

### Limitations

This review should be considered within the context of relevant limitations. Our search was restricted to a single database (PubMed) and to publications from the last five years, which aligned with our desire to review the recent literature but may have led to the exclusion of relevant studies indexed elsewhere or published earlier. We also relied on keyword searches without the use of MeSH terms, which may have affected retrieval precision. Although our search included the broader term “serious mental illness,” we did not explicitly include bipolar disorder as a search term, which may have limited the identification of some relevant studies focused on this population. As a narrative review, we did not conduct a formal quality assessment of included studies, and our synthesis therefore cannot distinguish the relative strength of evidence across sources. In addition, the literature we reviewed was predominately from the United States, Canada, and Europe, which limits generalizability to other healthcare systems, especially in non-Western countries. Finally, while our framework was not co-produced with patients or service users, many of the included studies incorporated first-person accounts, provider interviews, and qualitative perspectives, which informed our conclusions.

Of note, these limitations align with our primary aim of offering an interpretive synthesis (i.e., narrative review) rather than a systematic review that offers an exhaustive catalog of studies. We hoped to draw together diverse methodologies and highlight conceptual and systemic patterns, a particular strength of our study that may be less visible in more narrowly focused systematic reviews. Nonetheless, there remains an important role for more rigorous systematic and mixed method reviews that can build on this foundation by incorporating a broader range of databases, longer timeframes, quality assessments, and direct patient involvement.

## Figures and Tables

**Figure 1 healthcare-13-02387-f001:**
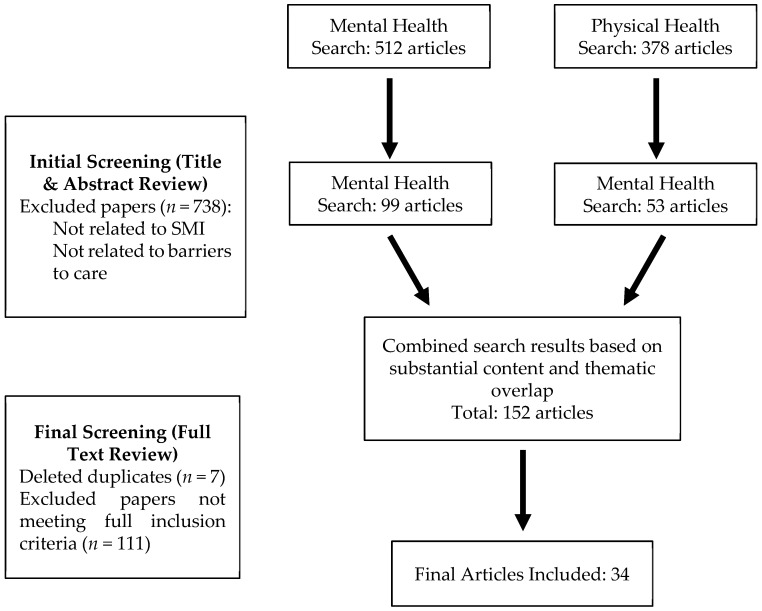
Search strategy.

**Table 1 healthcare-13-02387-t001:** Summary of results.

Authors and Year	Study Design	Sample Size	Location	Key Findings
Ambreen et al., 2025 [30]	Qualitative; semi-structured interviews	13 health administrators; 15 clinicians	Canada	Lack of integrated care was identified as a primary barrier to accessing healthcare for individuals with SMI.
Banerjee et al., 2021 [51]	Qualitative; semi-structured interviews	30 mothers with SMI	India	Barriers included perceived stigma, treatment side-effects, misinterpretations of information and health providers not having enough time.
Butler et al., 2020 [45]	Qualitative; semi-structured interviews	14 patients with SMI; 15 clinicians	UK	Clinician-reported barriers included lack of training, resource constraints, and uncertainty in their role.
Calderon et al., 2025 [43]	Qualitative; semi-structured interviews	Caregivers of 15 youth in high-risk-for-psychosis program	USA	Caregivers identified low mental health literacy, illness stigma, provider unavailability, and appropriateness and adequacy of referrals as barriers.
Carter et al., 2024 [48]	Mixed methods; survey and in-depth interviews	20 family physicians	Canada	Family physicians reported being unsure of availability of resources, varied comfort in recognizing psychosis, and a preference to co-manage psychosis.
Chea et al., 2024 [47]	Mixed methods; gap analysis and survey	40 stakeholders; 43 physicians	USA	Stakeholders reported stigma and fragmented resources as barriers. Physicians felt less prepared to manage SMI and reported lack of timely access, distance, and cost as barriers to care.
Cogley et al., 2023 [31]	Qualitative; semi-structured interviews	14 physical and 8 mental healthcare professionals	UK	The need for additional support, separation of physical and mental healthcare, limited confidence in working with SMI, and stigma were reported as barriers.
Colvin et al., 2024 [41]	Quantitative; cross-sectional survey	154 participants with SMI	USA	Elevated rates of medical mistrust in African American individuals with SMI represents a barrier to care.
Cunningham et al., 2023 [55]	Quantitative; online survey	335 individuals who had recently used primary care services	New Zealand	People with an SMI diagnosis had worse experiences across all quality measures; 20% reported diagnostic overshadowing.
Daneshvari et al., 2021 [35]	Quantitative; cross-sectional	271 individuals with SMI	USA	Higher scores on PANSS paranoid/belligerence were associated with delays in accessing care.
Grove et al., 2023 [25]	Qualitative; semi-structured interviews	50 clients of a community mental health center with SMI	USA	Convenience of vaccination location and access to free vaccination facilitated vaccine uptake. Fear and uncertainty were barriers.
Grünwald et al., 2021 [52]	Scoping review of medication reviews in SMI	55 articles	N/A	Low expectations of recovery, a perceived lack of capability to understand and participate in medication reviews, perceived risk of changing/stopping medication were barriers.
Ho et al., 2022 [54]	Integrative thematic review of carers perspectives	5 articles	N/A	Lack of coordination of care and diagnostic overshadowing were identified as barriers to care for individuals with SMI.
Jakobs et al., 2022 [33]	Qualitative; semi-structured interviews	13 general practitioners	The Netherlands	Barriers to cardiovascular risk screenings in SMI were underestimation of risk, burden on practitioners, poor information exchange, and skepticism about compliance.
Kohn et al., 2022 [39]	Qualitative; semi-structured interviews	18 healthcare professionals; 10 individuals with SMI	Belgium	Stigma, lack of communication, lack of training in SMI, organizational problems, and patient-related issues were identified as barriers to care.
Kozelka et al., 2024 [29]	Review	N/A	N/A	Access to a mobile device is not the same as digital literacy, and digital mental healthcare is not necessarily affordable for individuals with SMI.
Le Glaz et al., 2022 [53]	Systematic review	39 articles	N/A	Stigma and the desire for distance from people with SMI is high among medical students and physicians, which impacts medical care.
Lerbæk et al., 2021 [46]	Qualitative; interviews	Key informants on current practices of healthcare	Denmark	Negative attitudes and limited specialty knowledge were reported as barriers to healthcare for individuals with SMI.
Linz and Jermone-D’Emilia, 2024 [36]	Qualitative; interviews	15 women with SMI	USA	Barriers to breast cancer screenings included mental health symptoms, fear, distrust in the system, and lack of priority.
McKinley and Nienow, 2025 [44]	Qualitative; semi-structured interviews	36 veterans with SMI	USA	Barriers to accessing care included being unsure about when to access care, stigmatizing attitudes, and the complexity of the healthcare system.
Mishu et al., 2022 [32]	Qualitative; interviews	7 service users with SMI; 1 carer; 9 healthcare professionals	UK	Barriers to accessing oral healthcare included lack of communication, lack of support in visiting the dentist, payment, and long follow-up times.
Murphy et al., 2021 [38]	Mixed methods; semi-structured interviews	17 primary care physicians and 15 psychiatrists	USA	Clinicians identified lack of support, prioritization of other issues, communication, and other patient concerns as barriers to cancer screenings in SMI.
Myers et al., 2024 [28]	Qualitative; semi-structured interviews	18 individuals with SMI	USA	Monthly data limits and account management were identified as barriers to using Medicaid smartphones for healthcare.
Oluwoye et al., 2022 [24]	Quantitative	N/A	USA	Rural areas had a clear decrease in availability of coordinated specialty care programs, representing a geographic barrier to care.
Oluwoye et al., 2024 [23]	Quantitative; cross-sectional	225 service users with first-episode psychosis	USA	Longer commutes to coordinated specialty care clinics was associated with worse outcomes and lower rates of access, representing a geographic barrier to care.
Ratcliffe and Halpin, 2025 [56]	Review	N/A	UK	Viewing physical symptoms as psychosomatic and related stigma are barriers to care for COPD in SMI.
Reardon et al., 2024 [40]	Qualitative; semi-structured interviews	19 ACT team members	USA	Inadequate tools and training and limited patient awareness of physical care needs were identified as barriers to physical healthcare services.
Rosenfeld et al., 2022 [42]	Quantitative	N/A	USA	There is a significant mismatch between system demands and health literary skills of people with SMI, resulting in barriers to care.
Slanzi et al., 2024 [26]	Qualitative; focus groups	Stakeholders in rural US state	USA	Poverty and community mobility issues were identified as barriers to care.
Spooner et al., 2024 [34]	Qualitative; focus groups and interviews	20 people with SMI; 5 carers	Australia	Primary barriers to care were identified as impacts of mental illness, cognitive capacity, discrimination, and low income.
Williamson et al., 2025 [27]	Qualitative; interviews	14 stakeholders working with low-SES SMI individuals	USA	Lack of access to consistent internet and broken, lost, or uncharged phones were identified as barriers to telehealth care in SMI.
Woodall et al., 2024 [50]	Qualitative; semi-structured interviews	11 general practitioners; 8 psychiatrists; 11 managers/directors	UK	Competency gaps, lack of confidence in managing antipsychotics, communication, low expectations of individuals with SMI, and organizational issues were identified as barriers to psychiatric reviews in SMI.
Zatloff et al., 2020 [49]	Qualitative; semi-structured interviews	15 resident psychiatrists	USA	Perceived barriers to addressing reproductive health of women with SMI included lack of training, discomfort, and limited time.
Zhao and Mathis, 2024 [37]	Qualitative; semi-structured interviews	14 ACT clients with SMI; 7 clinicians from an ACT team	USA	Economic challenges, homelessness, and prioritization of mental health were identified as barriers to seeking physical healthcare.

**Table 2 healthcare-13-02387-t002:** Summary of themes.

Theme	Subtheme	Representative Studies
Structural and Logistical Barriers		
	Geographic location	[23,24,25,26]
	Access to technology and internet	[27,28,29]
	Disjointed medical and mental healthcare	[30,31,32,33]
Intrapersonal and Patient Level Barriers		
	Symptoms and psychological impacts of SMI	[31,32,33,34,35,36]
	Lack of awareness/prioritization of medical issues	[36,37,38,39,40]
	Medical mistrust	[31,36,41]
	Limited health literacy	[42,43,44]
Provider and System Level Barriers		
	Lack of knowledge/support for integrated care	[39,40,45,46]
	Lack of knowledge of SMI	[31,33,47,48,49,50]
	Stigma	[31,39,47,50,51,52,53]
	Diagnostic overshadowing	[30,53,54,55,56]

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
