# Peer review of "Behind the Gaps: A Narrative Review of Healthcare Barriers for Individuals with Serious Mental Illness"

_healthcare, 2025, doi:10.3390/healthcare13192387_

Round 1

Reviewer 1 Report

Comments and Suggestions for Authors

Thank you to the editors for the invitation to review this manuscript, and to the authors for their submission. The authors present a narrative review examining healthcare barriers faced by individuals with serious mental illness (SMI), synthesizing literature to identify three primary barrier categories: structural/logistical, intrapersonal/patient-level, and provider/system-level barriers. While the manuscript addresses an important healthcare disparity issue and provides a useful thematic synthesis, I have several substantial concerns that need to be addressed before this manuscript can be considered for publication in Healthcare. Below are my detailed comments:

  1. The manuscript title "Behind the Gaps: Understanding Healthcare Barriers for Individuals with Serious Mental Illness" should explicitly indicate that this is a narrative review.
  2. The introduction adequately establishes the mortality gap and health disparities, but lacks a clear justification for conducting another review when several systematic reviews on this topic already exist. What unique contribution does this narrative review offer beyond existing syntheses?
  3. The introduction section should incorporate recent advances in understanding healthcare barriers for SMI populations, particularly from global and rural perspectives. The authors should reference Zhang et al. (2024) "Barriers and facilitators to medical help-seeking in rural patients with mental illness: a qualitative Meta-synthesis" in Asian Nursing Research, which provides crucial insights into rural-specific challenges that complement the current urban-focused literature. Additionally, Chen and Zhou (2025) "Ensuring equitable access to mental health benefits for patients with severe mental illnesses in less developed rural areas in China" in Healthcare and Rehabilitation offers important perspectives on equity issues in resource-limited settings. These references would strengthen the rationale by highlighting both methodological advances (qualitative meta-synthesis approaches) and geographic gaps (non-Western, rural contexts) that the current review could address.
  4. The authors claim to complete a "narrative review" but provide insufficient justification for choosing this methodology over a systematic review or scoping review, which would be more rigorous and reproducible for this type of synthesis.
  5. In the Methods section, the search strategy is concerningly limited to only PubMed and restricted to the last five years. This narrow scope likely missed relevant studies from other databases (PsycINFO, CINAHL, EMBASE) and important foundational work published before 2019.
  6. The exclusion of 111 articles in the full-text review phase is substantial, yet the authors provide no detailed breakdown of exclusion reasons, which limits transparency and reproducibility.
  7. The search terms appear overly broad and lack Medical Subject Headings (MeSH) terms, potentially missing relevant studies while retrieving many irrelevant ones. The authors should provide the complete search string used.
  8. While the three-theme categorization is logical, there's significant overlap between categories (e.g., stigma appears in both patient-level and provider-level barriers) that isn't adequately addressed or theoretically justified.
  9. The quality assessment of included studies is completely absent. Without evaluating study quality, the review cannot distinguish between high-quality evidence and potentially biased or methodologically weak studies.
  10. Table 1 is useful but overly simplistic. It should include additional columns for study design, sample size, geographic location, and key findings to provide readers with more comprehensive information.
  11. The results section reads more like an annotated bibliography than a true synthesis. The authors should integrate findings across studies rather than summarizing each study sequentially.
  12. Critical gaps in the literature are not identified. For instance, there's minimal discussion of interventions that have successfully addressed these barriers, which would strengthen the practical implications.
  13. The discussion section's recommendations, while reasonable, lack evidence-based support from the reviewed literature. How do the authors know shared decision-making will address the identified barriers when this wasn't a focus of the reviewed studies?
  14. The geographic scope is predominantly US-focused (with some Canadian and European studies), limiting global applicability. This Western-centric bias should be acknowledged and discussed.
  15. The manuscript lacks discussion of intersectionality – how do these barriers differentially affect individuals based on race, gender, socioeconomic status, or other social determinants of health?
  16. Statistical synthesis or meta-analytical approaches are absent. While not always necessary for narrative reviews, some quantitative summary of barrier prevalence would strengthen the findings.
  17. The writing could be more concise. For example, the repeated listing of SMI conditions (schizophrenia, bipolar disorder, major depression) could be streamlined after the first definition.
  18. Several important barrier categories seem underdeveloped or missing, such as insurance/financial barriers, cultural barriers, and trauma-informed care considerations.
  19. The authors don't adequately address the bidirectional relationship between physical and mental health, which is crucial for understanding healthcare engagement in SMI populations.
  20. The search strategy diagram (Figure 1) contains an error – it shows different initial numbers than described in the text, raising concerns about accuracy and attention to detail.
  21. No patient or service user perspectives were included in developing the review framework or interpreting findings, which is a significant limitation given the topic's focus on patient barriers.
  22. The conclusion overstates the review's contributions without acknowledging its limitations or providing specific, actionable recommendations for different stakeholder groups (policymakers, clinicians, researchers).

Reviewer 2 Report

Comments and Suggestions for Authors

This paper is a narrative review of barriers to healthcare in persons with serious mental illness (SMI).

Though this topic has been addressed in several individual reports, the current paper is a worthy attempt to provide a thematic synthesis of the existing literature, and to use this as a basis for recommendations to improve accessibility to healthcare in persons with SMI.

There are previous reviews in this field, but with a more narrow focus (e.g., barriers to diabetes care or cancer treatment - Prathiksha et al., 2024; Glasdam et al., 2023) and this paper is, at least based on a preliminary literature review, the first to address this issue from a broader perspective.

There are certain aspects of the paper which would benefit from correction or clarification, as appropriate. These are enumerated below.

  1. It would be useful to mention the review type in the article's title, provided this is feasible and does not interfere with character limits or readability.
  2. In the introduction, it is stated that adherence to medication can improve some health outcomes (e.g., mortality in certain cases). While this is true, it is also worth mentioning the health hazards associated with long-term use of certain psychotropics (e.g., atypical antipsychotics with an adverse metabolic profile, divalproex, lithium). While the risks associated with these medications should not be overstated, they do exist, and they contribute significantly to medical morbidity in a subset of persons with SMI.
  3. Other reviews in this field (e.g., the two mentioned above, or the qualitative review by Chadwick et al. 2012) could be cited and discussed either in the Introduction or the Discussion, as appropriate. This would add important context and conceptual clarity to the current review.
  4. It is correctly stated that, as this is a narrative review, an exhaustive literature search is not required (lines 94-97). Nevertheless, there are certain guidelines / recommendations to improve the quality of narrative reviews (see, for example, Baethge et al., 2019). It would be helpful to refer to these, and to discuss how the current review aligns with them.
  5. Figure 1 contains some erroneous "markup" symbols. Please delete them and upload a corrected / "clean" version when submitting your revised manuscript.
  6. The search terms in lines 100-105 are unduly narrow. Bipolar disorders are also considered SMI and are associated with significant medical comorbidity and premature mortality. Why was the search confined to schizophrenia spectrum disorders?
  7. Much of the reviewed literature is derived from studies in developed countries with fairly good physical and mental healthcare infrastructure. Is the scenario different in developing countries, where mental health care is suboptimal and societal attitudes / stigma may be more of a problem? Did the authors observe any differences in barriers between research from high- and / low- or middle-income countries? As the current review aims to be broad / "global" in scope, these issues need to be addressed at least briefly.
  8. Possible interactions between the different levels / groups of barriers could be explored. It would also be worth examining if these barriers are worsened by other forms of social disadvantage (e.g., ethnic minority status, low income, unemployment, female gender in societies with higher gender inequality).
  9. The Discussion could include contrasts with some of the earlier, more specific reviews in this field, as mentioned in #3 above.
  10. The final paragraph of the Discussion could be converted into a final Conclusions section, providing a concise summary of the review's findings and their implications for practice and policy.

Reviewer 3 Report

Comments and Suggestions for Authors

Many thanks to the authors for exploring a significant and important topic related to mental health. The manuscript is very interesting, but I would like to make a few comments on the text.1

  • 1 Comment: The first MDPI journals usually use the American Chemical Society citation. Therefore, instead of: Serious mental illness (SMI) refers to a mental, behavioral, or emotional disorder that significantly interferes with an individual’s ability to function and carry out major life activities (Gonzalez et al., 2022; Substance Abuse and Mental Health Services Administration, 2024), it should be Serious mental illness (SMI) refers to a mental, behavioral, or emotional disorder that significantly interferes with an individual’s ability to function and carry out major life activities [1,2].
  • 2 Comment: In my opinion, the Aim of this literature review could be separated into its own section. For example, Section 1.2 could be titled "Aim of the Study" or "The Present Study." The first part of the introduction could be labeled as Section 1.1, "Workframe."
  • 3 Comment: The aim of the literature review seems quite broad. I suggest dividing the Aim into several parts and presenting them in sequence. For example: 1. To better understand the persistent mortality gap. 2. And so on.
  • 4 Comment: The manuscript does not specify the timeframe during which the article search was conducted.
  • 5 Comment: Why was a five-year period chosen as the exclusion criterion?
  • 6 Comment: The first figure has left out unnecessary characters (Microsoft Word characters that indicate spaces between words or the beginning of a new line) that can appear when formatting the figure.
  • 7 Comment: Were the PRISMA (Preferred Reporting Items for Systematic Reviews and Meta-Analyses) guidelines used in the literature review?
  • 8 Comment: In the first table, I recommend clearly distinguishing the Themes. It would help readers understand that there are three distinct Themes, each with its own subtopic. One option is to make the Themes bold, or you could consider rearranging the table structure for better clarity.
  • 9 Comment: In my opinion, the manuscript should include a prepared table that provides an overview of the selected articles, briefly describing the title, author(s), year, sampling method, and results.
  • 10 Comment: The manuscript is missing a limitations section.
  • 11 Comment: The manuscript is missing the Conclusion section.

I trust that my observations will provide you with valuable insights. Wishing you the best of success.

Round 2

Reviewer 1 Report

Comments and Suggestions for Authors

The authors have made substantial and constructive revisions in response to the initial review, and the manuscript has improved considerably compared to the original submission. I would, however, encourage the authors to further strengthen the integration of findings in the results section to move beyond study-by-study summaries and to provide a more cohesive thematic synthesis. 

Author Response

The authors have made substantial and constructive revisions in response to the initial review, and the manuscript has improved considerably compared to the original submission. I would, however, encourage the authors to further strengthen the integration of findings in the results section to move beyond study-by-study summaries and to provide a more cohesive thematic synthesis. 

We appreciate your thoughtful feedback and are grateful for the recognition of the revisions already undertaken. We understand the importance of presenting a more cohesive synthesis and have worked to further integrate findings in the results section, while still preserving the clarity of individual study contributions. We believe these revisions have improved integration and we hope the current version strikes the right balance between integration and readability. Of note, new changes are highlighted in green. Previous changes remain highlighted in yellow.

Thank you again! We truly appreciate the time and thought you have put into this review.

Reviewer 3 Report

Comments and Suggestions for Authors

I would like to thank the authors for taking my suggested revisions into consideration and for improving the manuscript accordingly. I wish them the utmost success in their future work.

Author Response

I would like to thank the authors for taking my suggested revisions into consideration and for improving the manuscript accordingly. I wish them the utmost success in their future work.

Thank you again. We truly appreciate the time and kind thoughtfulness that you put into your review.